# The Impact of Body Mass Index on Latent Tuberculosis Infection: Combined Assessment in People Living with HIV

**DOI:** 10.3390/pathogens14111078

**Published:** 2025-10-23

**Authors:** Jingxian Ning, Peng Lu, Yuchen Pan, Yilin Lian, Yu Zhang, Wenxin Jiang, Leonardo Martinez, Limei Zhu, Qiao Liu

**Affiliations:** 1Department of Epidemiology, School of Public Health, Nanjing Medical University, Nanjing 211166, China; ning329595@163.com (J.N.); lpjscdc@163.com (P.L.); wenxin001102@163.com (W.J.); 2Department of Chronic Communicable Disease, Jiangsu Provincial Center for Disease Control and Prevention, Nanjing 210009, China; panapan0011@163.com (Y.P.); yilin5522@163.com (Y.L.); zy173340@163.com (Y.Z.); 3Department of Epidemiology, School of Public Health, Southeast University, Nanjing 210009, China; 4Department of Epidemiology, School of Public Health, Boston University, Boston, MA 02118, USA; leomarti@bu.edu

**Keywords:** latent tuberculosis infection, people living with HIV, ESAT6-CFP10 skin test, body mass index

## Abstract

Background: Tuberculosis (TB) is a leading cause of death among people living with HIV (PLHIV). While body mass index (BMI) affects TB risk, its association with latent tuberculosis infection (LTBI) in PLHIV is unclear. High-transmission settings, such as prisons, may further increase LTBI risk, yet this relationship has not been studied across both prison and community populations of PLHIV. Methods: We conducted a dual cross-sectional study of PLHIV in Jiangsu Province, China, recruiting participants from a prison hospital in 2021 and community healthcare facilities from July to November 2023. BMI was calculated from measured height and weight. LTBI was identified by a positive ESAT6-CFP10 (EC) skin test or the QuantiFERON-TB Gold In-Tube (QFT-GIT) assay. Logistic regression and generalized additive models (GAMs) assessed the association between BMI and LTBI, adjusting for demographic, clinical, and behavioral confounders. Results: A total of 1799 PLHIV were included in the analysis, of whom 343 (19.07%) were recruited from prison settings and 1456 (80.93%) from community-based screening. The overall prevalence of LTBI was 13.79% (*n* = 248). Obesity (BMI ≥ 28 kg/m^2^) was linked to a significantly lower risk of LTBI (adjusted OR = 0.47, 95% CI: 0.23–0.95, *p* = 0.036), particularly when identified by EC testing (adjusted OR = 0.13, 95% CI: 0.03–0.54, *p* = 0.005). The BMI–LTBI association followed a nonlinear “U-shaped” pattern, with the lowest prevalence in individuals who were obese. Among those with CD4+ T cell counts < 500 cells/μL, the inverse association between obesity and LTBI was even more marked (adjusted OR = 0.20, 95% CI: 0.05–0.83, *p* = 0.027). Conclusion: In summary, obesity is significantly associated with a lower risk of LTBI among PLHIV, with an approximate 54% risk reduction. This inverse relationship was most pronounced when using the EC skin test.

## 1. Introduction

Tuberculosis (TB) remains the leading cause of morbidity and mortality among people living with HIV (PLHIV). In 2023, an estimated 39.9 million individuals were living with HIV globally, among whom approximately 161,000 deaths were attributable to TB coinfection [1]. Approximately one-quarter of the global population is infected with *Mycobacterium tuberculosis*. The intersection of HIV and LTBI represents a formidable public health challenge, especially in TB-endemic regions. Epidemiological evidence demonstrates that PLHIV face a 10- to 20-fold increased risk of TB reactivation compared to their HIV-negative counterparts [2]. HIV-induced immunosuppression compromises the host capacity to maintain *Mycobacterium tuberculosis* containment within granulomas, frequently resulting in extrapulmonary dissemination [2].

Several studies highlight a robust inverse association between body mass index (BMI) and Tuberculosis incidence and mortality. Elevated BMI, particularly overweight and obesity, has been linked to beneficial effects and reduced TB risk, with a 10–13.8% decrease in TB incidence for each unit increase in BMI across diverse settings [3,4,5,6,7]. Conversely, underweight status is associated with high mortality in TB patients. This inverse relationship is further supported by population-specific studies among PLHIV, where overweight and obese individuals exhibited significantly reduced risks of TB disease and all-cause mortality, while underweight status increased mortality [8,9,10]. However, the relationship between BMI and LTBI remains poorly characterized and controversial. While a cross-sectional study in the United States general population suggested a negative correlation between BMI and LTBI, identifying obesity as a potential protective [11] factor, conflicting evidence from other studies indicates that obesity may constitute a risk factor for LTBI [12]. This heterogeneity in findings across different populations and geographic settings underscores the need for clarification of this relationship.

Prisons act as significant “institutional amplifiers” of TB transmission globally, with incarcerated individuals experiencing a sharply elevated TB risk (over 1300 cases/100,000 person-years) compared to the general population [13]. Following release, former inmates face a 5.5-fold higher risk of active TB for up to seven years [13]. A study in Brazil indicated a dramatic increase in latent TB infection (LTBI) rates among inmates, rising from 9% at incarceration to an estimated 80% after five years [14]. As studies on BMI and LTBI in PLHIV in prisons are lacking, the aim of this study was to elucidate the association between BMI and LTBI among PLHIV. We conducted this investigation among PLHIV recruited from jail hospitals and community settings in Jiangsu Province, China, to address this critical knowledge gap in a middle-burden setting.

## 2. Methods

### 2.1. Study Population and Design

A dual cross-sectional study design was used to comprehensively evaluate the association between BMI and LTBI among PLHIV across diverse healthcare settings in China. The first cross-sectional survey was conducted in 2021 at a central prison hospital, leveraging a well-characterized study previously described in our earlier work [15]. This setting provided access to a population with potentially higher TB exposure risks due to congregate living conditions and healthcare disparities. The second survey, implemented between July and November 2023, employed a multi-site community-based approach across strategically selected healthcare facilities in Jiangsu Province, China. These facilities included tertiary hospitals in major urban centers (Nanjing, Wuxi, Changzhou, and Yangzhou) and secondary hospitals in smaller municipalities (Lishui District of Nanjing, Yixing City of Wuxi, Sheyang County of Yancheng, and Xiangshui County of Yancheng). This geographic diversity was designed to capture socioeconomic and demographic heterogeneity representative of the broader PLHIV population in eastern China.

Informed consent was obtained in accordance with the Declaration of Helsinki. Prior to enrollment, all participants underwent a comprehensive informed consent process administered by trained research personnel. The study was approved by the Ethics Committee of the Jiangsu Provincial Center for Disease Prevention and Controlwith ethics approval numbers JSJK2020-B008-01 and JSJK2023-B029-02.

### 2.2. Inclusion and Exclusion Criteria

Eligible participants met the following criteria: (1) a laboratory-confirmed HIV diagnosis; (2) completion of a standardized questionnaire; and (3) a valid QFT result (positive/negative). The exclusion criteria included: (1) missing CD4+ T-cell count or HIV viral load data; (2) indeterminate QFT results; (3) active TB (confirmed by chest radiography or clinical diagnosis); or (4) incomplete/invalid questionnaire responses.

### 2.3. Sample Size Calculation

The required sample size was estimated using the following formula:$$n=\frac{Z^{2}\cdot p\left(1-p\right)}{d^{2}},Z=1.96,d=0.05\;$$

Previous studies have reported that the prevalence of LTBI among PLHIV in China ranges from 20.8% to 64.3% [16]. Based on the formula, the estimated sample size ranged from 254 to 353 participants. After accounting for a 10% loss to follow-up, the required sample size was approximately 280–389 participants.

### 2.4. Procedures

Following established research protocols, certified research personnel conducted standardized interviews using validated, structured questionnaires to collect comprehensive demographic and clinical information, including age, sex, height, weight, ethnicity, BCG vaccination status, and history of incarceration. All study participants underwent a systematic chest radiographic examination before LTBI testing procedures to definitively exclude active tuberculosis disease, thereby ensuring an accurate classification of their latent infection status.

Blood sampling for QFT-GIT testing was performed according to the manufacturer’s specifications for sample collection, transport, and processing. Samples were incubated within 16 h of collection at 37 °C for 16–24 h, and interferon-γ levels were measured using an enzyme-linked immunosorbent assay. Subsequently, the EC skin test was administered intradermally (0.1 mL) on the volar aspect of the right forearm by trained healthcare personnel using standardised injection techniques. The EC test results were interpreted between 48 and 72 h post injection according to national guidelines. CD4 cell count and HIV viral load values were obtained through venous blood sampling as part of routine clinical care.

### 2.5. ESAT6-CFP10 Skin Test

The *Mycobacterium tuberculosis* specific antigen skin test (EC skin test) used in this study employed a recombinant fusion protein (ESAT6-CFP10) reagent (Yika^®^, Anhui Zhifei Longcom Biopharmaceutical Co., Ltd., Hefei, China), which has been approved for clinical use by the Chinese National Medical Products Administration. Each 0.1 mL dose contains 5 IU of recombinant ESAT6-CFP10 fusion protein.

The diagnostic accuracy and safety of this reagent have been validated through a series of rigorous multicenter, randomized, double-blind, parallel-controlled clinical trials [17,18,19,20,21]. Key clinical data demonstrated that, comparable to the T-SPOT.TB assay, the EC skin test yielded a diagnostic sensitivity of 90.64% for LTBI and a specificity of 88.20% in non-LTBI participants. Moreover, its specificity remained high (92.72%) among BCG-vaccinated participants, showing a significant improvement over the conventional tuberculin skin test [22]. The EC skin test has a favorable safety profile, with most adverse events being mild and self-limiting local or systemic reactions.

### 2.6. Diagnostic Interpretation and Definitions

PLHIV underwent the EC skin test administered on the volar aspect of the right forearm. Following each test administration, all subjects were monitored for a minimum of 30 min to identify any immediate adverse events. At 48 to 72 h post testing, the average transverse and longitudinal dimensions of induration and/or redness present at the administration sites were assessed. For the EC skin test, induration or redness ≥5 mm was considered a positive response [18].

The QFT-GIT protocol utilizes three distinct tubes: an antigen-coated tube with MTB-specific peptides, a mitogen tube as positive control, and a nil tube establishing the baseline. Interpretation criteria defined positive results as antigen-induced IFN-γ responses exceeding the nil baseline by ≥0.35 IU/mL while also demonstrating a ≥25% elevation relative to baseline values. Indeterminate findings occurred when nil tube IFN-γ concentrations surpassed 8.0 IU/mL or mitogen stimulation yielded responses < 0.5 IU/mL [23].

BMI was computed as body weight (kg) divided by height squared (m^2^) and categorized into four groups: underweight (<18.5), normal (18.5–23.9), overweight (24.0–27.9), and obese (≥28.0) [24].

### 2.7. Statistical Analysis

All data were double-entered into EpiData version 3.0 to minimise entry errors and analysed using SPSS version 26.0 (IBM Corp., Armonk, NY, USA). Baseline characteristics were compared using Pearson’s chi-square test or Fisher’s exact test for categorical variables, and non-parametric tests (Mann–Whitney U test or Kruskal–Wallis test) for continuous variables that were not normally distributed.

Univariate analyses were used to screen potential predictors of LTBI, including demographic variables, behavioral risk factors, incarceration history, BCG status, CD4 cell count, HIV viral load, and BMI categories. Variables with *p* < 0.10 in univariate analysis were entered into multivariate logistic regression models to estimate the adjusted odds ratios (ORs) and 95% confidence intervals (CIs) for the association between BMI and LTBI. Confounders were included in the final model based on significance (*p* < 0.05) in the multivariable analysis. All tests were two-sided, and a *p* < 0.05 was considered statistically significant. To further examine the potential nonlinear relationship between BMI and LTBI risk, generalised additive models (GAMs) with smoothing splines were fitted, adjusting for the same set of confounders identified in multivariable logistic regression.

## 3. Results

### 3.1. Characteristics of Participants by LTBI Status

This study included two cross-sectional surveys. In the 2021 Jiangsu prison survey, 353 PLHIV were recruited; after excluding 3 with active TB or abnormal chest X-rays, 3 with invalid questionnaires, and 16 with indeterminate QFT results, 343 participants were included. In the 2023 community hospital survey conducted across six cities in Jiangsu, 1506 PLHIV were recruited; after excluding 5 with active TB or abnormal chest X-rays, 36 without CD4 or viral load data, and 9 with indeterminate QFT results, 1456 participants were included. In total, 1799 PLHIV were eligible for the final analysis. The overall study flow is illustrated in Figure 1.

A total of 1799 PLHIV met the inclusion criteria and were included in the final analytical cohort. Among these participants, 1551 (86.21%) tested negative, and 248 (13.79%) tested positive for LTBI. The study population was predominantly male (89.77%), with a median age of 42 years (interquartile range, IQR: 33.5–54.0), and primarily of Han ethnicity (96.44%). The majority were recruited from the community (80.93%) and had a normal BMI (61.31%). Regarding educational attainment, 44.30% had completed junior secondary school or below, 23.40% had received secondary education, and 32.30% held a college degree or above (Table 1).

Of these, 343 (19.07%) were recruited from prison, while the remaining 1456 (80.93%) were identified through community-based screening (Appendix A). The overall prevalence of LTBI was 13.79% (*n* = 248). Among underweight, normal weight, overweight, and obese participants, the prevalence of LTBI was 10.38% (11/106), 14.23% (157/1103), 14.86% (70/471), and 8.4% (10/119), respectively. No significant differences in LTBI prevalence were observed across BMI categories. The BMI distribution of individuals grouped by LTBI status is shown in Figure 2.

### 3.2. Participant Characteristics by BMI Category

Overall, the LTBI group had a higher percentage of males and older individuals compared with the non-LTBI group across all BMI subgroups. The sociodemographic characteristics of the participants stratified by LTBI status and BMI categories are presented in Appendix A. In the univariate analysis, variables with *p* < 0.10 were considered potential risk factors for LTBI. These variables were subsequently included in multivariable logistic regression models to evaluate independent factors associated with LTBI within each BMI subgroup (Appendix A).

Among underweight participants, those with LTBI were significantly less likely to be unemployed or economically inactive compared to controls (aOR = 0.12, 95% CI: 0.02–0.58, *p* = 0.007). In the normal-weight subgroup, individuals with LTBI were significantly less likely to be of Han ethnicity (aOR = 0.38, 95% CI: 0.19–0.79, *p* = 0.01), to report no history of tuberculosis the contact (aOR = 0.38, 95% CI: 0.21–0.68, *p* = 0.001, to be recruited from community settings (aOR = 0.52, 95% CI: 0.33–0.80, *p* = 0.003, or to have attained lower educational levels (college or above vs. lowest educational level: aOR = 0.59, 95% CI: 0.37–0.94, *p* = 0.025), compared to controls. In the overweight subgroup, LTBI was associated with higher proportions of unemployment (aOR = 2.08, 95% CI: 1.16–3.72, *p* = 0.014) and female participants (aOR = 2.85, 95% CI: 1.28–6.34, *p* = 0.01), but a lower proportion of participants recruited from the community (aOR = 0.40, 95% CI: 0.19–0.83, *p* = 0.014). Among those classified as obese, the median age of LTBI-positive individuals was significantly lower than that of their LTBI-negative counterparts (aOR = 0.914, 95% CI: 0.842–0.992, *p* = 0.031), CD4+ T cell > 500 (aOR = 0.1, 95% CI: 0.04–0.99, *p* = 0.05).

### 3.3. Relationship Between BMI and LTBI

Table 2 presents the overall association between BMI and LTBI, using the normal weight group as the reference. After adjusting for age and sex, obesity remained inversely associated with LTBI (aOR = 0.47, 95% CI: 0.23–0.93, *p* = 0.030). This association persisted after further adjustment for potential confounders (aOR = 0.47, 95% CI: 0.23–0.95, *p* = 0.036), suggesting an inverse relationship between obesity and LTBI prevalence.

To further explore the nonlinear dose–response relationship between BMI and LTBI positivity, we employed a GAM (Figure 3). After adjusting for potential confounders, a significant nonlinear association was observed between BMI and LTBI positivity, as determined by EC skin testing (*p* = 0.0487). Specifically, the probability of EC positivity exhibited an approximate U-shaped pattern across the BMI spectrum, with the lowest positivity rate observed in individuals with a BMI range > 28 kg/m^2^ (obesity).

### 3.4. Relationship Between Obesity and LTBI Under CD4+ T Cell Stratification

We further investigated the association between obesity and LTBI within CD4+ T cell stratified subgroups. This study included 1799 people, of whom 1088 (60.48%) had a CD4+ T cell count ≤ 500, and 711 (39.52%) had a CD4+ T cell count > 500 (Appendix A).

Among individuals with CD4 counts ≤ 500 cells/μL, the prevalence of obesity was significantly lower in the LTBI group compared to the controls (*p* = 0.017) (Appendix A). As shown in Table 3, the multivariable analysis identified several protective factors against LTBI in this subgroup, including Han ethnicity (aOR = 0.34, 95% CI: 0.17–0.72; *p* = 0.004), higher education (aOR = 0.52, 95% CI: 0.31–0.87; *p* = 0.012), absence of TB contact history (aOR = 0.41, 95% CI: 0.22–0.75; *p* = 0.004), community-based recruitment (aOR = 0.42, 95% CI: 0.27–0.68; *p* < 0.001), and obesity (aOR = 0.20, 95% CI: 0.05–0.83; *p* = 0.027). Although a lower prevalence of obesity was also observed in the LTBI group within the CD4 counts > 500, this difference was not statistically significant.

### 3.5. Relationship Between BMI and LTBI Defined by EC Skin Test or QFT-GIT

Among 1799 PLHIV, the prevalence of LTBI was 8.06% according to EC skin test and 12.17% according to QFT assay. Stratified analyses revealed both protective and risk factors across BMI categories. In the normal-weight group, the protective factors consistently included an absence of TB contact history (EC: aOR = 0.32; QFT: aOR = 0.45), higher educational attainment (EC: aOR = 0.44; QFT: aOR = 0.60), and community-based recruitment (EC: aOR = 0.51; QFT: aOR = 0.41), with Han ethnicity showing additional protection in QFT (aOR = 0.34). Conversely, the long-term use of immunosuppressants (aOR = 0.41, EC) and higher CD4+ T cell counts (>500/μL, aOR = 3.81, QFT) were identified as risk factors. In the overweight group, LTBI was more prevalent among females (EC: aOR = 3.90; QFT: aOR = 2.85) and the unemployed (EC: aOR = 4.51), with lower positivity in community-based participants. For obese individuals, no significant predictors were found with the EC test, whereas older age (QFT: aOR = 1.09) was positively associated with LTBI according to QFT.

Obesity remained independently associated with a significantly reduced risk of LTBI when defined using the EC assay. In the unadjusted model, obesity was linked to an 83% reduction in LTBI risk (OR = 0.17, *p* = 0.013); this association persisted after adjustment for age and sex (aOR = 0.14, *p* = 0.006) and further adjustment for all covariates (aOR = 0.13, *p* = 0.005; Appendix A). Consistently, smaller skin induration diameters were observed in the overweight and obese groups compared with normal-weight individuals (Appendix A). In contrast, no significant association was found between obesity and LTBI based on the QFT results (aOR = 0.58, *p* = 0.134), although a weak positive correlation was detected between BMI and IFN-γ levels (Spearman’s r = 0.06, *p* = 0.019; Appendix A), suggesting limited biological relevance of BMI to IFN-γ variation.

## 4. Discussion

Through this comprehensive cross-sectional study focused on PLHIV, we discovered that obesity (BMI ≥ 28 kg/m^2^) was independently associated with a markedly reduced risk of LTBI, particularly when LTBI was identified via EC skin testing. A nonlinear U-shaped relationship between BMI and LTBI was observed, with the lowest prevalence at a BMI ≥ 28 kg/m^2^. To our knowledge, this is the first large-scale, multicenter investigation in China to evaluate the relationship between BMI and LTBI among PLHIV using both QFT and EC assays.

Our findings are broadly consistent with previous evidence from other countries reporting, an inverse associations between BMI and LTBI or active TB risk. Nguenha et al. conducted a randomized controlled trial of TB preventive therapy among PLHIV in South Africa, Mozambique, and Ethiopia. They reported that obesity was associated with a lower hazard of TB (aHR = 0.5, 95% CI 0.2–1.0). In the United States, another study found that underweight independently predicted higher mortality in adults with LTBI, whereas overweight and obesity showed no significant effect [25]. Similarly, Alaa Badawi and colleagues [11]. found that each one-unit increase in BMI corresponded to a 15% reduction in LTBI risk, and Mantri et al. [26] also reported a negative correlation between BMI and LTBI. In contrast, a study conducted in rural China found a positive association between obesity and QFT positivity (aOR = 1.17, 95% CI: 1.04–1.33); however, this analysis was limited to the general adult population and did not account for HIV status [12].

In our study, no significant association was observed between BMI and QFT positivity. The discrepancy between the EC skin test and QFT results may be attributed to their distinct immunological mechanisms and differential susceptibility to HIV-induced immunosuppression. QFT relies on a singular endpoint of IFN-γ secretion by T cells, which is significantly compromised in PLHIV individuals, leading to reduced sensitivity and increased false-negative rates, particularly in those with lower CD4+ counts [27,28,29,30,31]. In contrast, EC testing evaluates a complex delayed-type hypersensitivity response involving multi-faceted immunological cascades with both CD4+ and CD8+ T cell activation, demonstrating relative resilience to immune suppression [32]. This differential performance may explain why the inverse association between obesity and LTBI was detected by the EC skin test but not QFT.

Although the underlying mechanisms remain uncertain, emerging evidence suggests that obesity may influence LTBI risk through multiple immunological and metabolic pathways [33]. High BMI is characterised by elevated circulating levels of pro-inflammatory cytokines—such as IFN-γ, TNF-α, IL-22, IL-1α, IL-12, and GM-CSF—and concurrently reduced levels of anti-inflammatory cytokines, including IL-4, IL-5, and TGF-β [34]. This distinct immunoregulatory profile may underlie the enhanced containment of *Mycobacterium tuberculosis* and reduced progression to active disease in individuals with obesity [34]. Another plausible explanation is the immunomodulatory role of adipose tissue. Adipocytes and preadipocytes secrete numerous adipokines involved in both innate and adaptive immunity [35,36]. Leptin, secreted by adipocytes, plays a key role in T cell proliferation and activation. Its levels are higher in obese people and are associated with CD4+ T cell recovery and the suppression of HIV viral replication after antiretroviral treatment [37,38,39].

Moreover, recent evidence indicates that neutrophil-mediated immunity may provide an additional mechanistic link between BMI, inflammation, and LTBI risk. Neutrophils are a key component of innate host defense against *Mycobacterium tuberculosis* and their antimicrobial activity is largely mediated through the release of defensins and granulysin [40]. HIV infection is often accompanied by neutropenia [41], whereas obesity is associated with elevated neutrophil counts, higher C-reactive protein levels, and an increased neutrophil-to-lymphocyte ratio [42]. Recent studies have also demonstrated that an elevated neutrophil-to-lymphocyte ratio (NLR) or systemic immune-inflammation index (SII) are inversely associated with LTBI positivity [43,44]. Thus, elevated SII values likely reflect heightened neutrophil and platelet activity, contributing to the control of *Mycobacterium tuberculosis* growth and spread [43]. The interaction between obesity-related neutrophil activation and HIV-related immune suppression may therefore contribute to the inverse association observed between obesity and LTBI in our study.

BMI alone, however, may not fully capture body fat distribution. Visceral fat accumulation in PLHIV is linked to CD8+ T cell activation, insulin resistance, and higher levels of IL-6 and TNF-α, indicating that central obesity may play an important role in immune-metabolic interactions [45,46].

At the same time, CD4+ T-cell count remains a key determinant of TB risk in HIV infection, with declines strongly associated with increased TB incidence even in antiretroviral therapy (ART)-treated population [47]. The inverse association between obesity and LTBI was more evident among participants with CD4 counts < 500 cells/μL, suggesting that the relationship may vary with the level of immunosuppression. However, given the cross-sectional nature of this study, this inverse relationship does not establish causality, and reverse causation cannot be excluded. Prospective studies are needed to clarify whether higher BMI causally reduces LTBI risk or merely reflects preserved immune function.

Adipose tissue-derived adipokines, particularly leptin, may provide compensatory immunological benefits in the setting of moderate immunosuppression (CD4 < 500 cells/μL) by enhancing Th1 responses, promoting T cell proliferation, and facilitating mycobacterial containment through the metabolic reprogramming of immune cells [48,49,50,51]. In contrast, individuals with preserved CD4 counts (≥500 cells/μL) maintain robust antimycobacterial immunity independent of metabolic phenotype, rendering additional adipokine effects marginal. These findings suggest that integrating metabolic phenotypes with CD4 stratification could refine risk assessment for tuberculosis preventive therapy in PLHIV, with underweight individuals at CD4 < 500 cells/μL representing a particularly vulnerable subgroup requiring intensified intervention.

Prisons represent high-force-of-infection environments where overcrowding, poor ventilation, and close contact markedly increase the transmission risk, potentially overwhelming modest host-level protective effects of obesity-related immunometabolic factors [52,53]. Additionally, incarcerated populations exhibit distinct BMI distributions and nutritional profiles compared to those from community settings, which may further modify how obesity-related immunological effects manifest [54]. Future longitudinal research incorporating prison-specific exposure indicators alongside nutritional and immune parameters is warranted to elucidate the context-dependent nature of the BMI–LTBI relationship.

This study has several limitations. First, obesity in this study was assessed solely based on BMI, without considering other anthropometric indicators such as waist circumference or waist-to-hip ratio, which may better reflect central obesity and body fat distribution. Second, information on concomitant diseases was self-reported and not validated by medical records, which may reduce data accuracy and preclude further analysis of their potential effects on LTBI. Third, the cross-sectional nature of this study limits the ability to infer causality between BMI and LTBI. Therefore, prospective studies are warranted to explore the causality of BMI and LTBI. Fourth, our study lacks detailed longitudinal information on ART status, treatment duration, and HIV disease progression. Consequently, the potential impact of advanced HIV infection or AIDS-related disease on EC and QFT sensitivity cannot be fully assessed. Fifth, because this study was cross-sectional, variables such as duration of incarceration, physical activity, and chronic disease progression were not collected, which may influence BMI and partly explain the observed associations. Future prospective cohort studies are warranted to explore causal and time-dependent relationships between BMI dynamics and LTBI risk.

Despite these limitations, this study has several notable strengths. It included a large sample size across multiple regions with varying TB endemicity and is the firstmulticenter investigation in China to systematically evaluate the association between BMI and LTBI among PLHIV, utilising QFT, EC, and their combined diagnostic strategies. The findings provide significant empirical evidence of an inverse association between obesity and LTBI in this population, offering new insights and strategic considerations for improving targeted LTBI screening and preventive interventions in HIV care.

## 5. Conclusions

In summary, obesity (BMI ≥ 28 kg/m^2^) was significantly associated with an approximately 54% reduced risk of LTBI among PLHIV. This inverse association was most pronounced when LTBI was identified using the EC skin test. These findings highlight the importance of considering nutritional status in LTBI screening strategies. Weight status should be integrated into the identification of high-risk groups and guide the selection of appropriate diagnostic tools, contributing to more precise and tailored TB prevention strategies for PLHIV.

## Figures and Tables

**Figure 1 pathogens-14-01078-f001:**
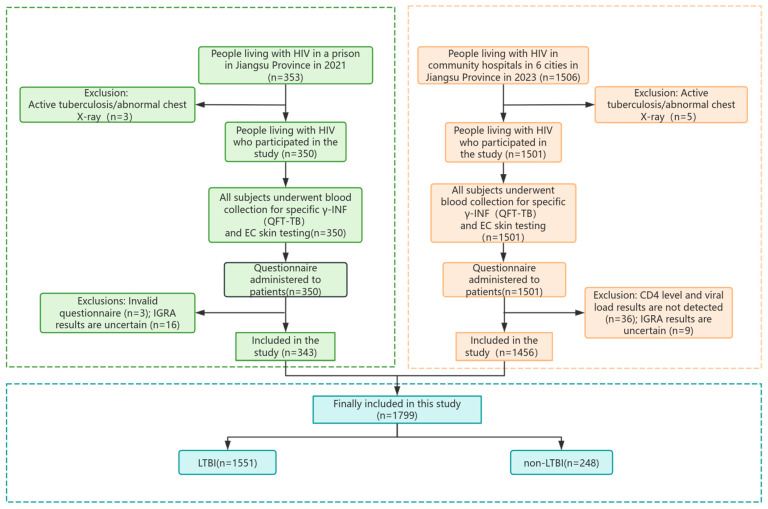
Flow chart of study inclusion and exclusion. Green boxes represent PLHIV in Jiangsu Province; orange boxes represent PLHIV in community hospitals across six cities in Jiangsu Province; blue boxes represent participants finally included in the study.

**Figure 2 pathogens-14-01078-f002:**
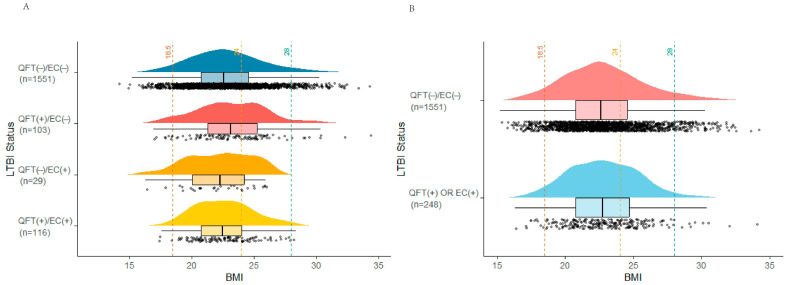
Distribution of BMI among individuals grouped by LTBI status based on QFT and/or EC skin test results. Panel (**A**) stratifies the study population into four groups based on QuantiFERON-TB Gold (QFT) and EC skin test (EC) results; Panel (**B**) simplifies the comparison to LTBI-negative (dual-negative) versus LTBI-positive (either positive) groups. The visualization combines density plots, box plots, and scatter plots to demonstrate data distribution characteristics. Vertical dashed lines indicate BMI thresholds for underweight (18.5), overweight (24), and obesity (28), respectively. Panel (**A**) blue represents QFT(−)/EC(−), red represents QFT(+)/EC(−), orange represents QFT(−)/EC(+), and yellow represents QFT(+)/EC(+); Panel (**B**) red represents QFT(−)/EC(−), and blue represents QFT(+) or EC(+).

**Figure 3 pathogens-14-01078-f003:**
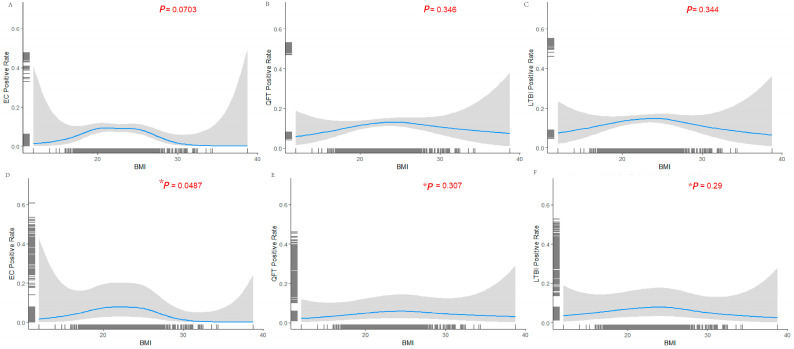
Generalized additive model analysis of the nonlinear association between BMI and LTBI risk among PLHIV. Association between BMI and positivity rates for EC, QFT, and LTBI status. Panels (**A**–**C**) show unadjusted associations, while Panels (**D**–**F**) present adjusted models controlling for potential confounders. Blue curves represent fitted trends, with shaded areas indicating 95% confidence intervals. Vertical tick marks denote individual BMI values. *p*-values (red) indicate the statistical significance of associations between BMI and test positivity. Results suggest a potential nonlinear relationship between BMI and LTBI-related test outcomes, with statistical significance observed only for the EC assay after adjustment. * After adjusting for confounding factors (Ethnicity, Education Level, Occupation, Contact history, CD4+ T cell count, Source, Gender, and Age), BMI was significantly associated with EC positivity (*p* = 0.0487).

**Table 1 pathogens-14-01078-t001:** Sociodemographic and clinical characteristics of PLHIV stratified by LTBI status (*n* = 1799).

Characteristic	Total (*n* = 1799, %)	Non-LTBI (*n* = 1551, %)	LTBI (*n* = 248, %)	Z/χ^2^	*p*
Age (years)	44.14 ± 13.51	44.23 ± 13.68	43.55 ± 12.39	0.80	0.425
Gender				0.67	0.412
Male	1615 (89.77)	1396 (90.01)	219 (88.31)		
Female	184 (10.23)	155 (9.99)	29 (11.69)		
Ethnic				50.13	<0.001
Minority	64 (3.56)	36 (2.32)	28 (11.29)		
Han	1735 (96.44)	1515 (97.68)	220 (88.71)		
HIV viral load (copies/mL)	43,520.61 ± 281,469.61	43,917.64 ± 287,887.74	41,037.57 ± 237,923.23	0.15	0.881
Education				19.78	<0.001
Primary school and below	797 (44.30)	659 (42.49)	138 (55.65)		
Middle and high school	421 (23.40)	363 (23.40)	58 (23.39)		
College degree or above	581 (32.30)	529 (34.11)	52 (20.97)		
Occupation				0.65	0.421
Incumbency	763 (42.41)	652 (42.04)	111 (44.76)		
Unemployed or retired	1036 (57.59)	899 (57.96)	137 (55.24)		
Smoke				15.24	<0.001
No	1115 (61.98)	989 (63.77)	126 (50.81)		
Yes	684 (38.02)	562 (36.23)	122 (49.19)		
Drink				4.40	0.036
No	1283 (71.32)	1120 (72.21)	163 (65.73)		
Yes	516 (28.68)	431 (27.79)	85 (34.27)		
Contact history				30.27	<0.001
Yes	108 (6.00)	74 (4.77)	34 (13.71)		
No	1691 (94.00)	1477 (95.23)	214 (86.29)		
BCG scars				7.58	0.006
No	609 (33.85)	506 (32.62)	103 (41.53)		
Yes	1190 (66.15)	1045 (67.38)	145 (58.47)		
CD4 T cell (cells/μL)				2.36	0.124
≤500	1088 (60.48)	949 (61.19)	139 (56.05)		
>500	711 (39.52)	602 (38.81)	109 (43.95)		
Source				60.60	<0.001
Prison	343 (19.07)	251 (16.18)	92 (37.10)		
Community	1456 (80.93)	1300 (83.82)	156 (62.90)		
BMI				4.58	0.205
Underweight	106 (5.89)	95 (6.13)	11 (4.44)		
Normal weight	1103 (61.31)	946 (60.99)	157 (63.31)		
Overweight	471 (26.18)	401 (25.85)	70 (28.23)		
Obese	119 (6.61)	109 (7.03)	10 (4.03)		
Diabetes				0.94	0.331
No	1431 (79.54)	1228 (79.17)	203 (81.85)		
Yes	368 (20.46)	323 (20.83)	45 (18.15)		
Cardiovascular disease				0.29	0.593
No	951 (52.86)	816 (52.61)	135 (54.44)		
Yes	848 (47.14)	735 (47.39)	113 (45.56)		
Silicosis				/	0.143 *
No	1794 (99.72)	1548 (99.81)	246 (99.19)		
Yes	5 (0.28)	3 (0.19)	2 (0.81)		
Nephropathy				1.88	0.170
No	1736 (96.50)	1493 (96.26)	243 (97.98)		
Yes	63 (3.50)	58 (3.74)	5 (2.02)		
Long-term use of immunosuppressants				2.03	0.154
No	1640 (91.16)	1408 (90.78)	232 (93.55)		
Yes	159 (8.84)	143 (9.22)	16 (6.45)		

* Fisher exact test.

**Table 2 pathogens-14-01078-t002:** Multivariable logistic regression analysis of factors associated with LTBI in PLHIV.

Variables	Model 1		Model 2		Model 3	
	cOR (95% CI)	*p*	aOR (95% CI)	*p*	aOR (95% CI)	*p*
BMI						
Normal weight	1.00 (Reference)		1.00 (Reference)		1.00 (Reference)	
Underweight	0.70 (0.37–1.33)	0.275	1.24 (0.79–1.94)	0.360	0.82 (0.39–1.73)	0.607
Overweight	1.05 (0.78–1.43)	0.745	0.00 (0.00–Inf)	0.997	1.09 (0.78–1.52)	0.627
Obese	0.55 (0.28–1.08)	0.082	0.47 (0.23–0.93)	0.030	0.48 (0.24–0.96)	0.038
Gender						
Males			1.00 (Reference)		1.00 (Reference)	
Females			1.24 (0.79–1.94)	0.360	0.99 (0.61–1.59)	0.956
Age (years)			0.00 (0.00–Inf)	0.997	0.00 (0.00–Inf)	0.997
Ethnic						
Minority					1.00 (Reference)	
Han					0.27 (0.15–0.50)	<0.001
Education						
Primary school and below					1.00 (Reference)	
Middle and high school					0.88 (0.60–1.28)	0.505
College degree or above					0.61 (0.40–0.94)	0.024
Occupation						
Incumbency					1.00 (Reference)	
Unemployed or retired					0.88 (0.65–1.18)	0.380
Contact history						
Yes					1.00 (Reference)	
No					0.43 (0.26–0.70)	<0.001
Source						
Prison					1.00 (Reference)	
Community					0.42 (0.29–0.61)	<0.001
CD4 T cell (cells/μL)						
≤500					1.00 (Reference)	
>500					1.58 (1.16–2.15)	0.003

Model 1: Crude model. Model 2: Adjusted for age and sex. Model 3: Adjusted for: Additionally adjusted for ethnicity, education level, occupation, contact history, CD4+ T cell count, and source. Reference: the baseline category against which other groups are compared in the regression analysis; Inf: abbreviation for “infinity,” indicating an infinite upper limit of the confidence interval when the estimate is undefined or extremely large.

**Table 3 pathogens-14-01078-t003:** Multivariable logistic regression analysis of risk factors for LTBI among PLHIV stratified by CD4+ T cell count.

Characteristic	PLHIV with CD4+ T Cell ≤ 500 (*n* = 1088)		PLHIV with CD4+ T Cell > 500 (*n* = 711)	
	cOR, 95CI, *p*	aOR, 95CI, *p*	cOR, 95CI, *p*	aOR, 95CI, *p*
Ethnic				
Minority	1.00 (Reference)	1.00 (Reference)	1.00 (Reference)	1.00 (Reference)
Han	0.15 (0.08–0.29), <0.001	0.34 (0.17–0.72), 0.004	0.25 (0.11–0.58), 0.001	0.37 (0.15–0.92), 0.034
Education				
Primary school and below	1.00 (Reference)	1.00 (Reference)	1.00 (Reference)	1.00 (Reference)
Middle and high school	0.64 (0.40–1.02), 0.06	0.75 (0.46–1.22), 0.253	0.86 (0.53–1.41), 0.556	1.17 (0.69–1.98), 0.562
College degree or above	0.39 (0.24–0.63), <0.001	0.52 (0.31–0.87), 0.013	0.52 (0.32–0.86), 0.01	0.73 (0.43–1.24), 0.24
Smoke				
No	1.00 (Reference)	1.00 (Reference)	/	/
Yes	2.17 (1.51–3.10), <0.001	1.22 (0.78–1.91), 0.374	/	/
Drink				
No	1.00 (Reference)	1.00 (Reference)	/	/
Yes	1.74 (1.20–2.52), 0.003	0.99 (0.63–1.54), 0.995	/	/
Contact history				
Yes	1.00 (Reference)	1.00 (Reference)	1.00 (Reference)	1.00 (Reference)
No	0.26 (0.15–0.45), <0.001	0.41 (0.22–0.75), 0.004	0.42 (0.21–0.86), 0.017	0.56 (0.26–1.20), 0.137
Source				
Prison	1.00 (Reference)	1.00 (Reference)	1.00 (Reference)	1.00 (Reference)
Community	0.27 (0.19–0.39), <0.001	0.43 (0.27–0.68), <0.001	0.37 (0.22–0.63), <0.001	0.43 (0.23–0.80), 0.007
Obese				
No	1.00 (Reference)	1.00 (Reference)	/	/
Yes	0.21 (0.05–0.86), 0.031	0.20 (0.05–0.83), 0.027	/	/
BCG scars				
No	1.00 (Reference)	1.00 (Reference)	1.00 (Reference)	1.00 (Reference)
Yes	0.69 (0.48–0.99), 0.046	0.94 (0.64–1.39), 0.760	0.61 (0.40–0.94), 0.026	0.87 (0.54–1.40), 0.56
HIV viral load (copies/mL)	/	/	1.00 (1.00–1.00), 0.074	1.00 (1.00–1.00), 0.085
Long-term use of immunosuppressants				
No	/	/	1.00 (Reference)	1.00 (Reference)
Yes	/	/	0.33 (0.10–1.07), 0.064	0.24 (0.07–0.82), 0.023

Reference: the baseline category against which other groups are compared in the regression analysis; “/”: indicates that the variable was not analyzed for that subgroup.

## Data Availability

The data dictionary can be made available upon request to the corresponding author.

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
