# Peer review of "The Impact of Body Mass Index on Latent Tuberculosis Infection: Combined Assessment in People Living with HIV"

_pathogens, 2025, doi:10.3390/pathogens14111078_

Round 1

Reviewer 1 Report

Comments and Suggestions for Authors

Major comments:

The authors have undertaken a very important investigation into the influence of body mass index on detection of latent M. tuberculosis infection in the general population and then separately a prison population living with HIV in a region in China. The following changes are required to presentation, design, analysis and discussion of the project:

Methods, Procedures: Line 111: The authors need to document the origin, specifications and validation specifics of the EC skin test with a literature reference documenting its validity.

The authors need to document any data on the neutrophil count of included patients. Neutrophil leucocytes are pivotal in immunity against M. tuberculosis infection (Martineau AR, Newton SM, Wilkinson KA, Kampmann B, Hall BM, Nawroly N, Packe GE, Davidson RN, Griffiths CJ, Wilkinson RJ. Neutrophil-mediated innate immune resistance to mycobacteria. J Clin Invest. 2007 Jul;117(7):1988-94). and a low neutrophil count is common in HIV infection: Levine AM, Karim R, Mack W, et al. Neutropenia in Human Immunodeficiency Virus Infection: Data From the Women's Interagency HIV Study. Arch Intern Med. 2006;166(4):405–410. doi:10.1001/archinte.166.4.405

Negative IGRA results were previously associated with a significantly higher neutrophil count in a Chinese population with tuberculosis and in a USA population with lack of latent M. tuberculosis infection: Zhang F, Gao Y, Li T, Zhang W. Development and validation of a nomogram for predicting false negative IGRA results in pulmonary tuberculosis patients using propensity score matching. PLoS One. 2025 Jul 2;20(7):e0327767. doi:10.1371/journal.pone.0327767, Pang T, Wang L, Zhang J, Duan S. Association between systemic immune-inflammation index and latent tuberculosis infection: a cross-sectional study. Front Med (Lausanne). 2025 Jul 30;12:1615302. doi: 10.3389/fmed.2025.1615302. 

Obesity is associated with a higher neutrophil count (Uribe-Querol E, Rosales C. Neutrophils Actively Contribute to Obesity-Associated Inflammation and Pathological Complications. Cells. 2022 Jun 10;11(12):1883. doi: 10.3390/cells11121883.), which may explain the findings of the authors.

Method section (line 143-145): The authors need to document a sample size calculation on which their study was based. The need to redo the multivariate logistic regression analysis and only include variables with p<0.05 in univariate analysis as only results statistically significant (the authors own definition of significance) variables should be entered.

Minor comments:

In line 48/49 (introduction) the sentence “Given that approximately one-quarter of the global population is infected 48 with Mycobacterium tuberculosis (Mtb)” is incomplete and the “Given that” should be deleted.

Table 2: Replace “Genger” by “Gender”

Comments on the Quality of English Language

In my minor comments I have mentioned the changes in language required.

Reviewer 2 Report

Comments and Suggestions for Authors

This dual cross-sectional study investigated the relationship between body mass index (BMI) and latent tuberculosis infection (LTBI) among 1,799 people living with HIV (PLHIV) in Jiangsu Province, China. Participants were recruited both from a prison hospital (19%) and community healthcare settings (81%). LTBI was defined using the ESAT6-CFP10 (EC) skin test and/or QuantiFERON-TB Gold In-Tube (QFT-GIT) assay.  Obesity was independently linked to a reduced risk of LTBI among PLHIV, with risk reductions up to ~54%. These findings suggest that nutritional status and BMI should be considered in LTBI screening and prevention strategies for HIV-infected populations, especially in high-transmission settings like prisons.  Some points to be adressed: 

  • The cross-sectional design prevents causal inference. Please emphasize more clearly that the observed inverse association between obesity and LTBI does not establish a protective effect, and reverse causality cannot be excluded.
  • A striking divergence was observed between EC and QFT results. While the EC test showed strong protective associations with obesity, QFT results did not. Please expand on the biological and immunological reasons for this discrepancy (e.g., immunosuppression impairing IFN-γ release in QFT, while DTH skin response may remain intact).
  • BMI was used as the only anthropometric marker. Central obesity or adipose tissue distribution may be more relevant to immune–metabolic interactions. Please acknowledge this limitation and suggest incorporation of waist circumference or body composition measures in future research.
  • Since prison settings showed higher LTBI prevalence, stratified results are valuable. Please discuss how incarceration-related transmission may modify or mask the BMI–LTBI association, and whether obesity’s apparent protective effect might be context-dependent.
  • The finding that the obesity–LTBI association was stronger in individuals with CD4 counts <500 is important. Expand on possible mechanisms (e.g., leptin-driven T-cell activation in immunosuppressed individuals) and discuss implications for preventive therapy policies.

Reviewer 3 Report

Comments and Suggestions for Authors

The association of nutritional quality and body weight with the risk of developing pulmonary tuberculosis is well known. In the current study, the authors analyzed the association of these indicators in patients with HIV.

Comments:

  1. How correct is it to combine different tests (EC-skin test and QuantiFERON-TB Gold In-Tube) into a single indicator "LTBI"? The abstract says that "LTBI was determined by a positive ESAT6-CFP10 (EC) skin test or the QuantiFERON-TB Gold In-Tube (QFT-GIT) assay." The methods describe that the tests were conducted sequentially: first the QFT, then the EC test. How could HIV infection affect the sensitivity of these tests?
  2. The study has a serious limitation – concomitant diseases were detected only according to the reports of the patients themselves, which greatly reduces the reliability of the data and makes it impossible to analyze the effect of concomitant diseases on pulmonary tuberculosis. It is unclear, for example, whether patients had COPD, and it is not clear what treatment they received for concomitant diseases. 
  3. Data on HIV infection is insufficient. The status of receiving antiretroviral therapy, its duration, and the length of the disease are not clear. People with a very low BMI in the HIV-positive cohort could have a more advanced stage of HIV infection (AIDS) with cachexia. They may have had false negative test results due to immune disorders.
  4. Many other data are unclear: whether the identified association of BMI with latent tuberculosis infection is permanent or age-dependent. It is not clear which came first - obesity or lack of physical therapy. The dynamics of body weight is not clear. This is especially important given that patients are in prison and their body weight may depend on the length of time in prison, the activation of chronic diseases associated with body weight.
  5. Why is the study timed out? In 2021, there was an active epidemic of coronavirus infection, which had a significant impact on concomitant diseases.

Round 2

Reviewer 1 Report

Comments and Suggestions for Authors

The authors have adequately addressed the comments.

Reviewer 3 Report

Comments and Suggestions for Authors

The authors answered my questions and made corrections to the text of the manuscript, which improved its quality. It is especially important to describe the limitations of the study.